# Multi-View Network Representation Learning Algorithm Research

**Zhonglin Ye** [1,2,3], **Haixing Zhao** [1,2,3,4,*], **Ke Zhang** [2,3,4] **and Yu Zhu** [2,3,4]

1 School of Computer Science, Shaanxi Normal University, Xi'an 710119, China; yzl2215@163.com
2 Tibetan Information Processing and Machine Translation Key Laboratory of Qinghai Province, Xining 810008, China; kezhang_qh@foxmail.com (K.Z.); zhuyu@qhu.edu.cn (Y.Z.)
3 Key Laboratory of Tibetan Information Processing, Ministry of Education, Xining 810008, China
4 School of Computer, Qinghai Normal University, Xining 810008, China
* Correspondence: h.x.zhao@163.com

**Abstract:** Network representation learning is a key research field in network data mining. In this paper, we propose a novel multi-view network representation algorithm (MVNR), which embeds multi-scale relations of network vertices into the low dimensional representation space. In contrast to existing approaches, MVNR explicitly encodes higher order information using $k$-step networks. In addition, we introduce the matrix forest index as a kind of network feature, which can be applied to balance the representation weights of different network views. We also research the relevance amongst MVNR and several excellent research achievements, including DeepWalk, node2vec and GraRep and so forth. We conduct our experiment on several real-world citation datasets and demonstrate that MVNR outperforms some new approaches using neural matrix factorization. Specifically, we demonstrate the efficiency of MVNR on network classification, visualization and link prediction tasks.

**Keywords:** network embedding; network representation; network embedding learning; network representation learning; network classification

---

## 1. Introduction

The network representation learning aims at learning and obtaining the low-dimensional, compressed and dense distributed representation vectors for various kinds of networks. It can be straightforwardly considered as the network encoding task for the networks, consequently, the nearest neighboring vertices have closer distance in the network representation space of lower dimension.

DeepWalk is the representative algorithm of network representation learning. However, DeepWalk [1] is able to incorporate higher-order network information by multiple steps of random walks. In this respect, WALKLETS [2] has conducted some positive explorations and researches and proved that multi-step random walk can encode higher-order features into the network representations. Meanwhile, high-order network representation learning can excavate valuable features only based on the existing network structural information in some sparse networks. In addition, the essence of DeepWalk based on Skip-Gram is to factorize the matrix of network structural features [3]. Fortunately, GraRep [4] and Network Embedding Update (NEU) [5] are proposed to learn the network representations, which can capture higher order network features. Thus, inspired by GraRep [4], WALKLETS [2] and NEU [5], we propose a joint learning framework of multi-view network representation learning algorithm (MVNR). MNVR aims to capture higher order structural relations into lower dimension embeddings by weighting $k$-step network representations of higher order into a single vector. The $k$-th step network is denoted as the $k$-th view

of the original network, therefore, MVNR is a network representation learning algorithm of integrating multi-view features.

The algorithm framework of MVNR is illustrated in Figure 1a,b.

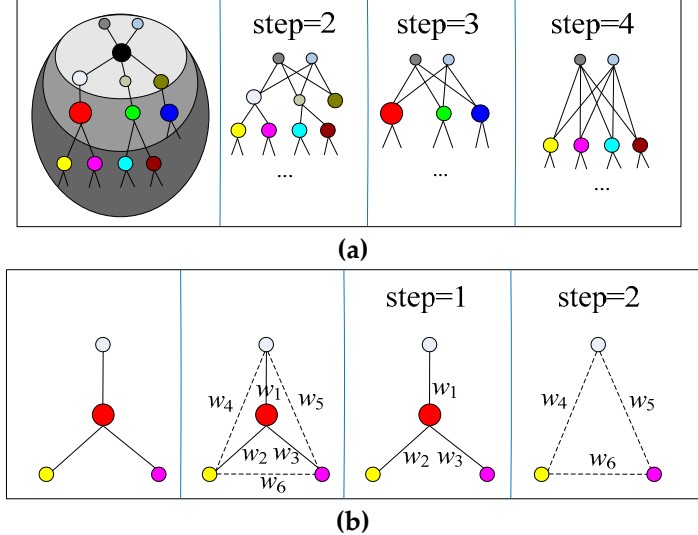

**(a)**

**(b)**

**Figure 1.** Algorithm framework of multi-view network representation (MVNR). (**a**) Higher order neighboring relations acquisition; (**b**) Edge weight calculation.

As shown in Figure 1a, MVNR can capture higher order structural relations, because we reconstruct *k*-step networks based on the length *k* of random walk on edges, the new network is different from the original network, the relations between of them are not the inclusive relations. In term of original network, the 2-step network is the 2nd order feature network when *k* is set as 2 and so on.

As shown in Figure 1b, the link weights consist of the certainty degrees of existing edges and the link probabilities of non-existing edges. The certainty degrees of existing edges are applied to 1-step network and the link probabilities of non-existing edges are used to *k*-step networks where *k* > 2.

Most previous works on network representation learning use a "one-size fits all" approach to train the learning model, where the single learnt representations are applied to various tasks, for example, DeepWalk [1], Tri-Party Deep Network Representation (TriDNR) [6] and node2vec [7] can only capture lower order features, which fails to explicitly capture higher order relations and global structural information. Especially in sparse networks, the existing network representation learning algorithms based on network structures are difficult to get valuable relations and structural features between vertices. GraRep [4] aims to factorize transition feature matrix of the network, which can explicitly encode the network features of higher order but it does not research how to weight the features form different order networks. WALKLETS [2] improves the procedure of random walk, which can capture the walk sequences by skipping pre-selected length of walk step. Nevertheless, WALKLETS [2] is still unable to obtain global features of the network, in fact, it is still the network representation learning algorithm based on local features of the network. NEU [5] can implicitly approximate higher order proximities with theoretical approximation bound. In fact, NEU [5] is more like an optimization algorithm, which optimizes the network representation obtained by DeepWalk [1], node2vec [7] and Line [8] and so forth. This kind of optimization process transforms the low-order network representations into the high-order network representations, however, NEU has no influence on modeling procedure of network representation learning.

We adopt a novel approach to jointly learn the network representations. For the final network representations, the different *k*-step networks should be given different weights. Therefore, we adopt the link probabilities of non-existing edges to weight different *k*-step network features. For the joint learning task, we use the inductive matrix completion algorithm to generate network representations

for each view. As the GraRep [4] algorithm, we concatenate different network representations from each view as the final representations. We believe that the *k*-step relations between different vertices reveal higher order relations and the global structural features, and it is essential to explicitly take full advantage of the different *k*-step network features for learning a better graph representation for various networks.

Our contributions are as follows:

(1) We introduce a novel network representation learning (MVNR), which explicitly captures higher order neighboring relations and global features. In network classification task, the performance of MVNR is superior to that of network representation algorithms based on the features of single view, such as DeepWalk, Line and node2vec. MVNR also outperforms the existing network representation learning algorithm based on the features of higher order, such as, GraRep and NEU.

(2) We introduce the Matrix Forest Index (MFI) to evaluate the weights of different *k*-step networks, which gives different weights for the representation vectors of different *k*-step networks in common representation vector space. This operation makes up for the important deficiency of GraRep algorithm. In addition, MFI features are calculated based on the network structure features. Therefore, it can provide sufficient structure features and improve the sparse problem of structure feature matrix in sparse networks.

(3) The visualization result of MVNR algorithm is better than that of node2vec and DeepWalk algorithm. It shows stronger cohesiveness and clearer classification boundary. Therefore, MVNR can learn the discriminative representation vectors. Consequently, it also shows excellent performance in link prediction tasks, and its link prediction performance is better than that of the baseline algorithms used in this paper.

## 2. Related Work

Network representation learning aims to embed the features of the network to representation spaces of lower dimension. The network features are mainly composed of structural features, text features of nodes, community attributes and labels, which is also the input of network representation learning. Network representation learning is similar to network coding task. The output of network representation learning is representation vector space, which makes some nodes with similar features have a closer distance in vector space.

Most network analytics approaches possess higher computation complexity. Network representation learning is an effective yet efficient method to deal with the network analytics challenges, which converts the network data into the representation space of lower dimension in which the network structural features and properties are maximumly preserved. Network representation learning benefits various network analytics tasks as the vector representations, the learnt representations also can be applied efficiently in both time and space. For example, we can conduct the node classification, node clustering, node recommendation, node retrieval, node ranking, link prediction and network visualization and so forth. There also exist other example scenarios, such as, multimedia network embedding, Information propagation and social networks alignment and so forth.

According to the application whether the network properties are applied to modeling procedure or not, the existing network representation learning algorithms are mainly divided into two kind of different research systems. The first kind of research system is only based on the network structural features and the second kind of research system is based on the joint learning model. Their representative algorithms are DeepWalk [1] and TriDNR [6], respectively.

The methods based on network structural features aim to explore and research the performance improvement of network representation learning based on the structure features of the network. For example, Perozzi et al. [1] proposed the DeepWalk algorithm in 2014. DeepWalk [1] adopts the Skip-Gram model in Word2Vec [9], modifying the input of the model from (Current Word, Context Words) pairs into (Current Vertex, Context Vertices) pairs. These pairs are inputted to a shadow neural network, which can fully embed the structure features into representation spaces of lower dimension.

Consequently, DeepWalk has been successfully applied to various tasks [10]. Line [8] proposed the first-order and second-order proximities function for learning large-scale network representation in 2015. Node2vec [7] improves random walk procedure of DeepWalk to sample neighboring vertices around the current vertex by introducing breadth-first search (BFS) and depth-first search (DFS) methods. Structural Deep Network Embedding (SDNE) [11] integrates global information during random walk procedure for network representation learning. Tu et al. [12] introduces the polysemy idea to representation learning which trains different network representations for current vertex according to different context nodes. There also exist some network representation approaches for the special network structures, such as DynamicTriad [13], DepthLGP [14] and Structural Deep Embedding for Hyper-Networks (DHNE) [15] and so forth. In addition, motivated by Generative Adversarial Networks (GAN) [16], some network representation learning algorithms introduce GAN to optimize the learning procedure, such as ANE [17], GraphGAN [18] and NetGan [19].

The method is based on the joint learning model and mainly improves the performance of network representation learning by means of introducing other attribute information of the network, such as community information, text contents and labels and so forth. According to existing literature findings, the performance of the joint network representation learning algorithm is better than that of the algorithm using only network structure information. Since the research foundation of DeepWalk, many researchers have proposed a variety of network representation learning algorithms based on joint learning model, such as, TriDNR [6], Content-Enhanced Network Embedding (CENE) [20], Graph Convolutional Network (GCN) [21], Discriminative Deep Random Walk (DDRW) [22], Planetoid [23], PTE [24], Modularized Nonnegative Matrix Factorization (M-NMF) [25], Accelerated Attributed Network Embedding (AANE) [26], COSINE [27] and Community Embedding (comE) [28] and so forth. Specifically, TriDNR [6] and CENE [20] incorporate the text contents into the procedure of network representation learning by treating the content information as a special kind of node. M-NMF [25] is a modularized nonnegative matrix factorization approach, which incorporates the community structures into network representations. Both COSINE [27] and comE [28] also incorporate the community information into network representations. AANE [26] is an accelerated attributed network embedding approach, which enables the joint learning framework to be conducted by decomposing the complicated learning and training into many sub-problems. In addition, based on the fact of matrix decomposition of DeepWalk [1], Yang et al. [29] proposes the Text-Associated Deep Walk (TADW) algorithm which ensembles text contents into matrix factorization algorithm. Based on the TADW [29] algorithm, the Max-Margin Deep Walk (MMDW) [30] algorithm adopts maximum margin algorithm to optimize the learnt representation vectors. There exist some community preserving [31] and heterogeneous [32] network representation algorithms.

## 3. Our Method

In this paper, we propose a novel network representation algorithm, MVNR, to learn multi-view representations of vertices in a network. Firstly, we introduce the multi-view strategy to define the $k$-step network features. Meanwhile, we introduce link prediction index to evaluate the link weights for existing and non-existing edges. Based on the link weights, we then propose a new approach to calculate the corresponding weights for each view's representations. Finally, we propose a new approach to jointly learn the representations based on the different views and weight information. Consequently, our model integrates rich local structural information associated with the network, capturing the global structural properties of the network.

### 3.1. Formalization

Suppose that $G = (V, E)$ is a network, we first denote some parameters, $V$ denotes the node set, $E$ denotes the edge set and $E \in V \times V$. $|V|$ is the size of node set. For each node $v \in V$, the purpose of network representation aims to learn the low-dimensional representation $r_v \in \mathbb{R}^k$, where $k$ is the length of the representation vector and $k$ is less than $|V|$. In addition, $r_v \in \mathbb{R}^k$ is not only applied to

network classification task but also can be used for various machine learning tasks, such as clustering, link prediction, recommendation systems and so forth.

## 3.2. Feature Extraction for Different k-Step Networks

DeepWalk uses Skip-Gram model for big-scale network representation learning, which captures context vertices by random walk algorithm. The objective of the DeepWalk is to maximize the following average log probability:

$$\frac{1}{L}\sum_{i=1}^{L}\sum_{-t\leq j\leq t, j\neq 0}\log p(v_{j+i}|v_i), \tag{1}$$

where $L$ denotes the number of vertices, $v_i$ denotes the current vertex and $v_{j+i}$ denotes the context vertices of the current vertex $v_i$. $t$ is the number of context vertices before and after the current vertex $v_i$. $t$ indicates the context window size to be $2t + 1$. Moreover, the conditional probability $p(v_{j+i}|v_i)$ can be defined by $p(v_{i+j}|v_i) = \exp(o_{v_{i+j}}^T r_{v_i})/\sum_{v\in V}\exp(o_v^T r_{v_i})$, where $r_v$ and $o_v$ is the input and output latent variable, namely, the input and output representation vectors of $v$.

Based on DeepWalk and PageRank, the research achievements from TADW [29] show that DeepWalk aims to factorize the following matrix:

$$M_{ij} = \log\frac{[e_i(Pr + Pr^2 + \cdots + Pr^t)]_j}{t}, \tag{2}$$

where $Pr_{ij} = 1/d_i$, if $(i,j) \in E$, and $Pr_{ij} = 0$ otherwise. $d_i$ is the degree value of vertex $i$.

As shown in Equation (3), matrix $M$ captures the $t$-th step neighboring vertices. Node2vec adopts BFS and DFS to capture higher order neighboring vertices, GraRep captures higher order network feature matrix by the $k$-step probability transition matrix $B^k$, and $B^k$ is the multiplication form of multiple $B$, where $B = D^{-1}A$, $A$ is an adjacency matrix, $D$ is a diagonal matrix. One shortage of GraRep is that the inverse matrix $D^{-1}$ does not exist when the network is sparse. Based on the Equation (2), TADW [29] presents a simplified target matrix and finds a balance between speed and accuracy. Consequently, matrix $M$ can be formulated as follows:

$$M = \frac{Pr + Pr^2}{2} \tag{3}$$

Matrix $M$ captures the first-order and second-order neighbors for vertices with small computation complexity.

GraRep uses the matrix $(D^{-1}A)^k$ to represent higher order feature matrix. In MVNR, we capture higher order features by defining the adjacency matrix $A^{(k)}$ as GraRep, but we perform some optimization operations for the matrix $A^{(k)}$. Unlike GraRep algorithm, $A^{(k)}$ is the adjacency matrix of $k$-step network. In GraRep algorithm, $(D^{-1}A)^k$ can be regarded as the variation form of $A^k$. $A^{(k)}$ can be regarded as the reachability matrix within $k$ steps. Therefore, the optimization operations of $A^{(k)}$ adopt a similar form as GraRep, but the essence of matrix $A^{(k)}$ and $A^k$ is very different. Moreover, the structure feature matrix constructions based on $A^{(k)}$ and $A^k$ are also very different. These differences are the optimization operations of $A^{(k)}$ in this paper.

Here, we first adopt $A^{(k)}$ to denote the adjacency matrix of $k$-step network and $A^k = [a_{ij}^k]$.

Matrix $A^{(k)}$ can be formulated as follows:

$$A^{(k)} = [a_{ij}^{(k)}] \tag{4}$$

where $A^{(k)}$ is the adjacency matrix of the $k$-step network, $a_{ij}^{(k)}$ is the matrix elements of matrix $A^{(k)}$ and adjacency matrix $A$ consists of 0 and 1, so we set the element value as 0 or 1 in $A^{(k)}$, thus $a_{ij}^{(k)} = 0$, if $a_{ij}^k = 0$, and $a_{ij}^{(k)} = 1$ otherwise, where $a_{ij}^k$ is the matrix elements of matrix $A^k$. The Equation (4) is different from the

transition matrix of GraRep, for each $k$-step network, we reconstruct the $k$-step network based on matrix $A^k$ and $A^{(k)}$. In fact, $A^{(k)}$ is the probability of reachability.

For each step network, we can get the network structural features as follows:

$$M^{(k)} = \frac{A^{(k)} + (A^{(k)})^2}{2}. \tag{5}$$

Here, $A^{(k)}$ is the adjacency matrix of the $k$-th network, which is different from the probability transition matrix $Pr$ in Equation (3). The Equation (5) and Equation (3) have the same form, but they contain different elements. $A^{(k)}$ only consists of element 0 and 1. $Pr_{ij} = 1/d_i$, if $(i,j) \in E$, and $Pr_{ij} = 0$ otherwise. $d_i$ is the degree value of vertex $i$. Both transition probability matrix $Pr$ and adjacency matrix $A^{(k)}$ are the structure feature matrices of the network. Moreover, GraRep factorizes the transition probability matrix based on adjacency matrix and achieves better network representation performance. The factorization objective matrix of SGNS algorithm is the Equation (2), but the TADW algorithm factorizes the matrix $M$ in Equation (3) simplified by Equation (2) and achieves excellent network representation learning performance. Therefore, we replace the transition probability matrix $Pr$ in Equation (3) with the adjacency matrix $A^{(k)}$. On the one hand, it is based on the above comparative analysis, on the other hand, it is based on such consideration that the factorization of adjacency matrix has lower computational complexity.

Therefore, matrix $M^{(k)}$ can be regarded as the structural features of $k$-step network, which is essentially different from the feature matrix of GraRep. The structure features of GraRep is as follows:

$$Y_{i,j}^k = \log(\frac{B_{i,j}^k}{\sum_t B_{t,j}^k}) - \log(\beta), \quad \beta = 1/|V|. \tag{6}$$

Here, $|V|$ is the number of vertices in graph $G$, $B_{i,j}^k$ is the element from $i$-th row and $j$-th column of the matrix $B^k$, and $B = D^{-1}A$.

### 3.3. Feature Weighting for Different k-Step Networks

For the different $k$-step networks, the same vertex pairs should be given the different weight values. However, DeepWalk, GraRep and node2vec neglect the weight information for higher order neighbors. Therefore, the proposed MVNR in this paper solves this problem by following procedures.

We first introduce Matrix Forest Index (MFI) to evaluate the weights between vertices for 1-step network. For the weights between vertices on $k$-step network. We compare the performance of the MFI algorithm with some classical link prediction algorithms, such as the algorithms based on common neighbor, the algorithms based on random walk and the algorithms based on path. We find that MFI achieves the best link prediction performance on several real citation network datasets. In addition, MFI can be calculated only through the Laplacian matrix of the network. The input of Laplacian matrix is the adjacency matrix of the network. The input matrix is exactly the same as that of the proposed MVNR algorithm. Unlike other link prediction algorithms, the input, calculation procedure and result of MFI can be embedded in the learning framework of the MVNR algorithm. Therefore, we choose the MFI algorithm to measure the weights between vertices.

MFI is formulated as follows:

$$S = (I + L)^{-1} \tag{7}$$

Here, $S = [s_{ij}]$ is a matrix which is constructed by the matrix forest index. $L$ is the Laplacian matrix of $G$, $I$ is an identity matrix. $L$ can be calculated based on adjacency matrix, its detailed calculation method can be found in the Algorithm 1.

We do not use MFI algorithm and adjacency matrix of $k$-step to calculate the weights between vertices in $k$-step network. We calculate the weights between vertices in $k$-step network by using the weights between vertices in 1-step network and the adjacency matrix of $k$-step network. Because the weights calculated by MFI in 1-step network include the weights of existing edges and the future

connection probabilities of non-existing edges between vertices. Only through such calculation can the weights of different *k*-step networks be different and hierarchical and also play a role in adjusting the weights of different *k*-step network representations. For the *k*-step network, we define its weight matrix as follows:

$$W^{(k)} = [w_{ij}^{(k)}], \quad w_{ij}^{(k)} = s_{ij} \times a_{ij}^{(k)}, \tag{8}$$

where $a_{ij}^{(k)} = \begin{cases} 0, & a_{ij}^{k} = 0 \\ 1, & a_{ij}^{k} \geq 1 \end{cases}$, $s_{ij} \in S, a_{ij}^{(k)} \in A^{(k)}$. $a_{ij}^{(k)}$ is the element of the adjacency matrix $A^{(k)}$.

As mentioned above, the link weights consist of the certainty degrees of existing edges and the link probabilities of non-existing edges. The certainty degrees of existing edges are applied to 1-step network and the link probabilities of non-existing edges are used to *k*-step networks where *k* > 2. Through the Equations (7) and (8), we find that we only compute the value of MFI for one time to different *k*-step networks and then through the MFI matrix, we construct the weight matrices of different *k*-step networks. For the 1-step network, we only retain the similarity values between two vertices with one edge in the MFI matrix and delete the similarity values between two vertices without any edge in the MFI matrix. Therefore, we define the similarity value between two vertices with one edge as the certainty degrees of existing edges. For the 2-step network, we only retain the similarity values between two vertices, where these two vertices are reachable within two steps and the similarity value is the future link probabilities of non-existing edges of the 1-step network. Specifically, the weights of the 2-step network are the similarity values between the current central vertex and the neighbor's neighboring vertices. Therefore, we regard this kind of similarity value as the future link probabilities between two vertices without any edge. The certainty degrees of existing edges and the link probabilities of non-existing edges are calculated by the MFI matrix of 1-step network. The specific calculation process is shown in the Equation (8) and Algorithm 1.

For each *k*-step network, we reconstruct the *k*-step weight matrix based on matrix *S* and matrix $A^{(k)}$. We only retain the link weights of existing edges and neglect the weights of non-existing edges between vertices. The weight matrix factor is not only to balance *k*-step network representation but also it can be regarded as network weight features, which can be integrated into network representations.

In the *k*-step network, the edge relationship has been established between vertices, where the edge relationship does not exist in the original network. Weight matrix of *k*-step network can not only balance network representation vectors of different *k*-step networks but also can be regarded as network weight features of the network, which can be integrated into network representation framework. Specifically, we construct the edge weight matrix of *k*-step network and the weight matrix of *k*-step network is different. The weights of the edges of the original network are larger. The edge relationship of the 2-step network is the reachability relationship within 2 step between two vertices in the original network, which can be set as 1 and 0. By MFI similarity calculation, the edge weights of 2-step network are the similarity values between the current vertex and the neighbor's neighboring vertices in the original network. Therefore, the weights in the 2-step network is less than that in the 1-step network. So, the higher the order of the network, the smaller the edge weight in weight matrix is. By using the hierarchical weight matrix, we can combine the weight matrix with the network structure feature matrix for joint network representation learning. Thus, different weight factors are assigned to different *k*-step network representation vectors by weight matrix.

### 3.4. Joint Learning of MVNR

Suppose that matrix $M \in \mathbb{R}^{m \times n}$ admits an approximation of low rank *k*, where $k \ll \{m, n\}$. Based on the matrix factorization, Yu et al. [33] propose a matrix factorization approach with a penalty term constraint, which aims to find $X \in \mathbb{R}^{k \times n}$ and $Y \in \mathbb{R}^{k \times m}$ and minimize the likelihood

$$\min_{W,H} \|M - (X^T Y)\|_F^2 + \frac{\alpha}{2}(\|X\|_F^2 + \|Y\|_F^2), \tag{9}$$

where $\alpha$ is a harmonic factor to balance two components in Equation (9). Specifically, Equation (9) aims at factorizing $M \in \mathbb{R}^{m \times n}$ into two matrices $X \in \mathbb{R}^{k \times m}$ and $Y \in \mathbb{R}^{k \times n}$, where $M \approx XY$. Here, matrix $M \in \mathbb{R}^{m \times n}$ can be regarded as the feature matrix of the network $G$ for the task of network representation learning. Matrix $X^T \in \mathbb{R}^{n \times k}$ can be regarded as the learnt representation matrix of the network $G$. We do not use the model proposed by Equation (9) in MVNR algorithm.

We use the Inductive Matrix Completion (IMC) algorithm presented by Natarajan and Dhillon [34], which adopts two known matrices to factorize matrix $M$ as follows:

$$\min_{W,H} \sum_{(i,j)\in\Omega} \left( M_{ij} - (P^T X^T Y Q)_{ij} \right)^2 + \frac{\beta}{2} (\| X \|_F^2 + \| Y \|_F^2), \tag{10}$$

where $\Omega$ denotes the sample set of matrix $M \in \mathbb{R}^{m \times n}$, here, $P \in \mathbb{R}^{p \times m}$ and $Q \in \mathbb{R}^{q \times n}$ are two known feature matrices. $\beta$ is the hyper-parameter to balance two components $M_{ij} - (P^T X^T Y Q)_{ij}$ and $\| X \|_F^2 + \| Y \|_F^2$. IMC aim at learning the matrix $X \in \mathbb{R}^{d \times p}$ and matrix $Y \in \mathbb{R}^{d \times q}$ to meet $M \approx P^T X^T Y Q$. The Equation (10) is originally applied to complete gene-disease matrix using gene and disease feature matrix. Motivated by IMC, we integrate the link weights of existing edges into Equation (10), the purpose of MVNR is to solve matrices $X$ and $Y$ to minimize the objective function as follows:

$$\min_{W,H} \| M^{(k)} - ((X^{(k)})^T Y^{(k)} W^{(k)}) \|_F^2 + \frac{\lambda}{2} (\|X^{(k)}\|_F^2 + \|Y^{(k)}\|_F^2). \tag{11}$$

Here, $W^{(k)} \in \mathbb{R}^{q \times n}$ is the weight matrix of the $k$-th step network, $\lambda$ is the hyper-parameter to balance two components $M^{(k)} - ((X^{(k)})^T Y^{(k)} W^{(k)})$ and $\|X^{(k)}\|_F^2 + \|Y^{(k)}\|_F^2$. For the Equation (10), we denote the matrix $P \in \mathbb{R}^{p \times m}$ as the identity matrix $E \in \mathbb{R}^{p \times m}$. Consequently, we get the Equation (11) based on Equation (10). For each $k$-step network, we denote $R^{(k)} = (X^{(k)})^T$ as representation matrix of the $k$-th network. The final representation vectors can be concatenated as follows:

$$R = R^{(1)} \oplus R^{(2)} \oplus \cdots \oplus R^{(k)}, \tag{12}$$

where $k = \{1, 2, \ldots, K\}$ and $K$ is a pre-selected constant. As Equation (12), we concatenate all $k$-step representations to form the global representations, which can be used in various machine learning tasks.

We introduce matrix $W^{(k)}$ to adjust the weights of the different $k$-step networks and remedy the defect of ignoring the weight information for GraRep. For the $k$-step network representation $r^{(k)}$, we apply L2 Norm to normalize the learnt network representations which can show better performance in some evaluation tasks, such as network classification and link prediction.

We give the algorithm description of MVNR in Algorithm 1.

*3.5. Complexity Analysis*

In the proposed MVNR, the training procedure can be divided into the following parts: $M$ construction, $M$ factorization using IMC and representations concatenation. In matrix $M$, the number of rows equals the number of columns, meanwhile, the number of rows and columns equals the number of vertices in the network, which is defined as $|V|$. Thus, the time complexity of constructing matrix $M$ is $O(|V|^3)$.

For the Equation (11), we introduce the optimization approach proposed by Yu et al. [33]. For each $k$-step network, we use the known weight matrix $W$ to factorize matrix $M$ of network feature based on IMC algorithm. The output of the Equation (11) is the matrices $X$ and $Y$, where $M \approx XYW$. Therefore, the time complexity of each iteration of minimizing $X$ and $Y$ is $O(\text{nnz}(M)d + |V|dq + |V|d^2)$ in Equation (10), where nnz $(M)$ denotes the number of non-zero elements in $M$, $d$ denotes the vector length of the network representations. The time complexity of the representations concatenation is $O(|V|kd)$.

---

**Algorithm 1** MVNR Algorithm.

---

**Input:**

    Adjacency matrix $A$ on network.

    Maximum transition step $K$.

    Harmonic factor $\lambda$.

    Dimension of representation vector $d$.

**Output:**

    Matrix of the network representation $R$.

**Content:**

1.     Calculate $A = 1/d_{ij}$.

2.     Calculate matrix forest index $S = [s_{ij}]$:

        $I$ = sparse(GetIdentityMatrix (RowSize($A$))).

        $D = I$.

        % The positions of 1 in Identity Matrix are set to the degree of vertex as follows:

          degree of vertex as follows:

        $D$(ConvertTologicMatrix ($D$)) = RowSum($A$).

        $L = D - A$.

        $S$ = inv($I + L$). % inv is to computes the inverse.

    **for** $k = 1$ to $K$

       (1)     Get $k$-step transition matrix $A^{(k)} = [a_{ij}^{(k)}]$:

           1.1     Compute $A^1, A^2, A^3, \ldots, A^K$.

           1.2     Get the matrix $A^{(k)}$:

$$A^{(k)} = [a_{ij}^{(k)}]$$

          where $a_{ij}^{(k)} = \begin{cases} 0, & a_{ij}^k = 0 \\ 1, & a_{ij}^k \geq 1 \end{cases}$.

       (2)     Get the network structural feature matrix $M^{(k)}$:

$$M^{(k)} = (A^{(k)} + (A^{(k)})^2)/2.$$

       (3)     Get the weight matrix $W^{(k)}$:

$$W^{(k)} = [w_{ij}^{(k)}], \ w_{ij}^{(k)} = s_{ij} \times a_{ij}^{(k)}.$$

       (4)     Construct the representation vector $R^{(k)} = (X^{(k)})^T$:

$$\min_{W,H} \|M^{(k)} - ((X^{(k)})^T Y^{(k)} W^{(k)})\|_F^2 + \frac{\lambda}{2}(\|X^{(k)}\|_F^2 + \|Y^{(k)}\|_F^2).$$

    **end for**

3.     Concatenate all the $k$-step representations:

$$R = R^{(1)} \oplus R^{(2)} \oplus \cdots \oplus R^{(k)}.$$

---

## 4. Experiments and Evaluations

In our works, we conduct our experiment on three real-world citation networks and evaluate the performance of MVNR based on network classification, visualization and link prediction.

## 4.1. Datasets Setup

We conduct our experiments on Citeseer (https://github.com/albertyang33/TADW), Cora (https://relational.fit.cvut.cz/dataset/CORA) and Wiki (https://github.com/albertyang33/TADW), which are not weighted networks. Wiki [29,30,35] is much denser than Cora and Citeseer. The dataset descriptions are as follows (Table 1):

**Table 1.** Network datasets descriptions.

| Dataset | Node | Edge | The Number of Classes | Average Degree | Average Path Length |
|---------|------|------|-----------------------|----------------|---------------------|
| Citeseer | 3312 | 4732 | 6 | 2.857 | 9.036 |
| Cora | 2708 | 5429 | 7 | 4.01 | 6.31 |
| Wiki | 2405 | 17981 | 19 | 14.953 | 3.65 |

## 4.2. Baseline Algorithms

To evaluate the performance of MVNR, we introduce the following algorithms of network representation as baseline algorithms. The representation length of all the baseline methods is set to 200.

**DeepWalk:** DeepWalk is the most classical algorithm which adopts only network structural features to learn the network representations. we use the Skip-Gram model and Hierarchical Softmax for DeepWalk. **Line**: Line is a recently proposed algorithm to solve the representation learning of big-scale networks. Line is a method capturing local features based on the probability loss function, which provides 1st Line and 2nd Line models. Similar as our MVNR, it uses 1-setp and 2-step neighboring vertices to learn the network representations. **node2vec**: node2vec also adopts the BFS and DFS strategy to capture the higher order neighboring vertices. Meanwhile, node2vec define a second-order random walk to balance the DFS and BFS. **NEU**: This algorithm is from the paper "Fast Network Embedding Enhancement via High Order Proximity Approximation," which is a higher order network representation learning algorithm. In this paper, the NEU algorithm convert the lower-order representations trained by DeepWalk into the higher-order representation.

**GraRep**: It is a new network representation learning algorithm, which can capture higher order network features by matrix factorization and representation concatenation.

**Weight-based Matrix Factorization (WMF)**: For the $k$-step network, we define the matrix $W^{(k)}$ as the link weight matrix of the $k$-th network view. We then use the SVD algorithm to factorize the matrix $W^{(k)}$, here $W^{(k)} \approx U^{(k)} S^{(k)} (V^{(k)})^T$. We regard $U^{(k)} (S^{(k)})^{0.5}$ as the network representation vectors.

**MVNR:** It is our proposed algorithm. In this experiment, we adopt the different $k$ values to capture the $k$-step vertex relational information and learn the network representation. $\lambda$ as 0.5 for Citeseer and Cora and 0.1 for Wiki.

**Unweighted MVNR**: This algorithm is a simplified MVNR algorithm with the equal weights for different $k$-step networks, we call this algorithm as Un_MVNR. Other optimizations are the same as MVNR.

The MVNR algorithm proposed is a higher order network representation learning algorithm. Therefore, we choose three classical network representation learning algorithms of lower order, such as DeepWalk, Line and node2vec. node2vec improves the random walk of DeepWalk, Line and DeepWalk aim to factorize the structure feature matrix of network; the proposed MVNR is also an NRL algorithm based on matrix factorization form of DeepWalk. So, we take DeepWalk, Line and node2vec as the baseline algorithms. In addition, we also choose two classical network representation learning algorithms of higher order for comparison and analysis, such as GraRep and NEU. DeepWalk is the most classical network representation learning algorithm. Line and node2vec are classical network representation learning algorithms based on DeepWalk. GraRep and NEU are higher order network representation learning algorithms accepted by top-level conference and they are also classical

representative algorithms of higher order representation learning algorithms. WMF and Un_MVNR are the simplified versions of MVNR algorithm, which are used to measure the performance influence of weight matrix on MVNR algorithm. The parameter setting of these algorithms is introduced in algorithm description part of Section 4.2.

The parameter settings for the selected baseline models are shown in Table 2:

**Table 2.** Parameter settings.

|  | Random Walk Length | Random Walk Number | Representation Vector Size | The Number of Orders | Context Window Size |
|---|---|---|---|---|---|
| DeepWalk | 80 | 10 | 200 | - | 5 |
| Line | - | - | 200 | 2 | 5 |
| node2vec | 80 | 10 | 200 | - | 5 |
| NEU | 80 | 10 | 200 | - | 5 |
| GraRep | - | - | 100 for $k$-th network | 1,3,6 | - |
| WMF | - | - | 100 | 1 | - |
| MVNR | - | - | 100 for $k$-th network | 1,3,6 | - |

### 4.3. Multi-Class Classification

In this experiment, we first learn the network representations based on MVNR algorithm. Based on the learnt representations, we then train a Support Vector Machine (SVM) classifier with different proportions of training sets. For the training set, we let training proportions of dataset vary from 10% to 90%. The remaining data is regarded as the testing set. We repeat the procedure for 10 times and report the average accuracy value. We set the $K$ value as 1, 3 and 6. For example, we concatenate the network representations of the 1-step, 2-step and 3-step networks when $K$ is 3. The detailed results are shown in Tables 3–5.

**Table 3.** Accuracy (%) of vertex classification on Citeseer.

| %Labeled Nodes | 10% | 20% | 30% | 40% | 50% | 60% | 70% | 80% | 90% |
|---|---|---|---|---|---|---|---|---|---|
| DeepWalk | 48.31 | 50.36 | 51.33 | 52.31 | 52.85 | 53.33 | 52.98 | 53.47 | 53.71 |
| Line | 39.82 | 46.83 | 49.02 | 50.65 | 53.77 | 54.20 | 53.87 | 54.67 | 53.82 |
| node2vec | 54.38 | 57.29 | 58.64 | 59.53 | 59.63 | 59.88 | 60.43 | 61.36 | 62.42 |
| WMF ($K = 1$) | 52.14 | 54.17 | 55.38 | 55.42 | 56.41 | 56.57 | 57.23 | 57.79 | 57.22 |
| NEU | 50.23 | 52.00 | 53.62 | 54.07 | 54.46 | 53.74 | 54.34 | 55.51 | 55.29 |
| GraRep ($K = 1$) | 26.21 | 34.13 | 38.30 | 40.82 | 41.92 | 44.68 | 44.29 | 45.30 | 44.83 |
| GraRep ($K = 3$) | 45.07 | 50.95 | 53.40 | 54.21 | 54.87 | 55.75 | 55.54 | 55.15 | 54.22 |
| GraRep ($K = 6$) | 50.72 | 53.02 | 54.21 | 55.22 | 55.51 | 56.17 | 55.99 | 55.78 | 57.94 |
| Un_MVNR ($K = 1$) | 51.07 | 55.65 | 56.96 | 57.84 | 58.04 | 58.30 | 58.95 | 58.21 | 58.76 |
| Un_MVNR ($K = 3$) | 54.54 | 58.36 | 59.17 | 60.90 | 61.43 | 61.76 | 61.67 | 62.38 | 62.32 |
| Un_MVNR ($K = 6$) | 54.90 | 59.23 | 60.31 | 61.12 | 61.96 | 62.47 | 62.49 | 63.15 | 62.14 |
| MVNR ($K = 1$) | 54.32 | 56.67 | 57.99 | 58.94 | 59.26 | 59.59 | 60.11 | 59.89 | 60.39 |
| MVNR ($K = 3$) | 56.99 | 59.80 | 61.57 | 62.54 | 63.73 | 64.71 | 64.93 | 64.98 | 66.22 |
| MVNR ($K = 6$) | 57.63 | 60.61 | 61.39 | 63.69 | 64.15 | 65.05 | 65.59 | 65.97 | 66.97 |

**Table 4.** Accuracy (%) of vertex classification on Cora.

| %Labeled Nodes | 10% | 20% | 30% | 40% | 50% | 60% | 70% | 80% | 90% |
|---|---|---|---|---|---|---|---|---|---|
| DeepWalk | 73.29 | 75.46 | 76.19 | 77.49 | 77.89 | 77.83 | 78.86 | 79.05 | 78.62 |
| Line | 65.13 | 70.17 | 72.20 | 72.92 | 73.45 | 75.67 | 75.25 | 76.78 | 79.34 |
| node2vec | 76.30 | 79.26 | 80.43 | 80.70 | 81.13 | 81.26 | 82.18 | 81.63 | 82.81 |
| WMF ($K = 1$) | 68.77 | 71.75 | 72.53 | 73.68 | 73.71 | 74.54 | 74.56 | 75.04 | 75.55 |
| NEU | 73.21 | 76.23 | 77.49 | 78.25 | 78.89 | 79.94 | 80.85 | 81.00 | 81.04 |
| GraRep ($K = 1$) | 63.92 | 72.51 | 74.85 | 75.92 | 76.80 | 76.74 | 76.85 | 77.19 | 77.92 |
| GraRep ($K = 3$) | 72.60 | 77.34 | 78.34 | 79.39 | 79.43 | 80.30 | 80.32 | 80.68 | 79.88 |
| GraRep ($K = 6$) | 75.95 | 78.54 | 79.64 | 80.40 | 80.70 | 81.42 | 81.74 | 81.36 | 82.92 |
| Un_MVNR ($K = 1$) | 70.43 | 76.97 | 78.61 | 79.63 | 80.68 | 80.31 | 80.16 | 81.60 | 82.25 |
| Un_MVNR ($K = 3$) | 75.32 | 79.18 | 80.06 | 80.88 | 81.44 | 81.62 | 82.46 | 82.25 | 83.59 |
| Un_MVNR ($K = 6$) | 76.15 | 78.52 | 79.98 | 80.65 | 81.70 | 82.35 | 82.98 | 81.97 | 82.59 |
| MVNR ($K = 1$) | 74.96 | 77.86 | 78.69 | 80.04 | 80.26 | 80.69 | 80.91 | 80.99 | 81.81 |
| MVNR ($K = 3$) | 77.50 | 79.69 | 81.15 | 81.42 | 82.38 | 82.86 | 83.05 | 83.79 | 82.78 |
| MVNR ($K = 6$) | 78.24 | 80.44 | 81.08 | 82.08 | 82.29 | 83.12 | 83.25 | 83.90 | 83.81 |

**Table 5.** Accuracy (%) of vertex classification on Wiki.

| %Labeled Nodes | 10% | 20% | 30% | 40% | 50% | 60% | 70% | 80% | 90% |
|---|---|---|---|---|---|---|---|---|---|
| DeepWalk | 52.15 | 58.79 | 61.69 | 62.80 | 62.92 | 63.05 | 64.38 | 64.25 | 64.41 |
| Line | 51.17 | 53.62 | 57.81 | 57.26 | 58.94 | 62.46 | 62.24 | 66.74 | 67.35 |
| node2vec | 57.51 | 61.11 | 63.17 | 63.73 | 64.39 | 65.40 | 65.58 | 65.37 | 66.50 |
| WMF ($K = 1$) | 35.22 | 39.68 | 40.92 | 42.71 | 42.97 | 43.95 | 46.04 | 44.31 | 44.87 |
| NEU | 52.03 | 58.36 | 60.08 | 61.95 | 62.98 | 63.90 | 63.16 | 64.56 | 64.08 |
| GraRep ($K = 1$) | 55.50 | 59.24 | 60.40 | 61.30 | 61.43 | 62.11 | 63.28 | 61.08 | 62.95 |
| GraRep ($K = 3$) | 55.98 | 61.15 | 62.79 | 63.75 | 64.21 | 65.24 | 65.20 | 65.08 | 65.54 |
| GraRep ($K = 6$) | 56.73 | 60.42 | 62.77 | 64.43 | 65.42 | 65.44 | 65.96 | 66.97 | 66.37 |
| Un_MVNR ($K = 1$) | 57.22 | 61.88 | 63.19 | 65.68 | 65.03 | 66.11 | 66.61 | 65.89 | 68.91 |
| Un_MVNR ($K = 3$) | 55.63 | 60.84 | 63.76 | 64.65 | 65.41 | 66.07 | 67.11 | 66.70 | 67.70 |
| Un_MVNR ($K = 6$) | 54.72 | 60.49 | 62.31 | 63.63 | 64.40 | 65.41 | 65.65 | 66.91 | 67.95 |
| MVNR ($K = 1$) | 57.74 | 62.27 | 64.26 | 64.51 | 65.81 | 66.50 | 66.69 | 67.93 | 68.33 |
| MVNR ($K = 3$) | 58.29 | 61.98 | 63.81 | 65.56 | 66.08 | 66.54 | 66.62 | 67.92 | 68.08 |
| MVNR ($K = 6$) | 57.55 | 62.37 | 63.99 | 65.92 | 66.41 | 66.74 | 67.03 | 68.75 | 69.50 |

In classification evaluation, MVNR consistently and significantly improves the accuracy of diverse network representations on three evaluation datasets. In addition, MVNR outperforms the baseline algorithms and showing its excellent performance with diverse training ratios on Citeseer, Cora and Wiki datasets. Obviously, the performance of higher order network representation learning (GraRep, NEU and MVNR) is better than that of the network representation learning algorithm based on local feature acquisition (DeepWalk, Line and node2vec).

In this classification evaluation, we set *K* as 1, 3 and 6, which means that we learn and concatenate the network representations of the 1-step, {1, 2, 3}-step and {1, 2, 3, 4, 5, 6}-step networks for GraRep and MVNR. The experimental results show that although GraRep can capture neighboring relations of higher order, MVNR appears to capture better neighboring vertices of higher order than GraRep on three datasets. On the other hand, we use the NEU algorithm to transform the learnt network representations generated by DeepWalk algorithm into higher order one but the performance of the network representations of higher order is inferior than that of the proposed MVNR algorithm in network classification task. The main reason is that GraRep ignores the weight information between different *k*-step networks and it also neglects the edge weights in each *k*-step network. Importantly, MVNR gives corresponding weight value to edges of each *k*-step network, which can indirectly affect the weights between different *k*-step networks. NEU is not involved in the learning procedure of

high-order network representation, and NEU is only a high-order transformation of the network representations trained by other network representation algorithms. NEU [6] is essentially different from GraRep and MVNR. MVNR can embed the global information into lower dimension embeddings by explicitly constructing the global feature matrix of the network. In addition, MVNR also can learn the network representations of the 2nd, 3rd and 4th order by constructing $k$-step networks.

DeepWalk and node2vec belong to this kind of algorithms based on local feature acquisition. Based on DeepWalk, node2vectakes the micro-view and macro-view of the vertices into account. Line-provides a network representation learning algorithm for various large-scale networks. Although Line-improves the training speed of network representation learning, the precision of network representation learning is not as good as DeepWalk and node2vec. Because Line only considers first order similarity and second order similarity of the vertices. The proposed MVNR in this paper fully considers the global information of the vertices by explicitly constructing the global feature matrix. Thus, the performance of MVNR algorithm is superior to DeepWalk, node2vec and Line. GraRep and NEU encode the features of higher order into the representation space of lower dimension. GraRep does not consider the weights of edges, it also does not consider the weights between different $k$-step networks. NEU only performs higher order transformation of network representation for the learnt network representations trained by other network representation learning algorithms. The MVNR algorithm proposed in this paper gives corresponding weights to the edges of different $k$-step networks. The larger the $k$, the smaller the weights of the edges, which indirectly gives different weights to different $k$-step networks. In addition, the MVNR algorithm explicitly embeds higher order features of the network, global information and weight information into the representation space through the joint learning model. Therefore, the performance of the MVNR algorithm is better than that of GraRep, and NEU algorithms in Tables 3–5.

In addition, the experimental results show that the classification performance of MVNR is better than that of Un_MVNR on Citeseer, Cora and Wiki datasets. This demonstrates that adding different weight information to different $k$-step networks can improve the performance of network representation learning. In fact, weight information also plays another role, namely, another feature view of the network structure. Therefore, it can also make up for the problem of sparse network structure.

In Section 3.5, we theoretically discuss the time complexity of the proposed MVNR algorithm. In addition, we provide the empirical result comparing running time performance with GraRep and NEU algorithms on Citeseer dataset. First, we installed a virtual machine on a notebook computer and then we installed the MVNR running environment on that virtual machine. The notebook's memory is 8G, the processor is Core I3 2.53 GHz and the memory allocated to the virtual machine is 3G. Experimental results show that MVNR takes 15 minutes and 01 second to train the model of network representation learning, GraRep takes 5 min and 35 s to train the model of network representation learning when K is set to 6 and NEU takes about 7 seconds to convert the learnt network representations trained by DeepWalk. Although MVNR takes the longest time, the network classification performance of MVNR is superior to the GraRep and NEU algorithms on Citeseer, Cora and Wiki datasets. The main reason why MVNR become time-consuming is that MVNR uses weight matrix $W$ to factorize the matrix $M$ based on IMC algorithm, GraRep purely uses SVD to factorize the feature matrix of network, and NEU is just a multiplication and addition operation for the learnt representations.

### 4.4. Parameter Sensitivity

MVNR has two parameters needed to be adjusted, they are the representation length (vector dimension) and K. Here, K is the pre-selected constant for the $k$-step network, where $1 \leq k \leq K$. We let K value range from 1 to 6 and the representation length of $k$-step network varies within 25, 50, 100 and 200 for Citeseer, Cora and Wiki. The last network representation size 75, 150, 300 and 600. We fix the training ratio to 0.5 and we then test classification accuracy with diverse representation lengths and K values in Figure 2.

As shown in Figure 2, the accuracy of network classification shows an increasingly trend with the increase of *K* and representation length. Importantly, the growth of accuracy becomes slower with the growth of the representation vector length. In addition, the average degree of Cora is bigger than that of Citeseer and Wiki, so that the accuracy fluctuation of Cora within a small range compared with Citeseer and Wiki.

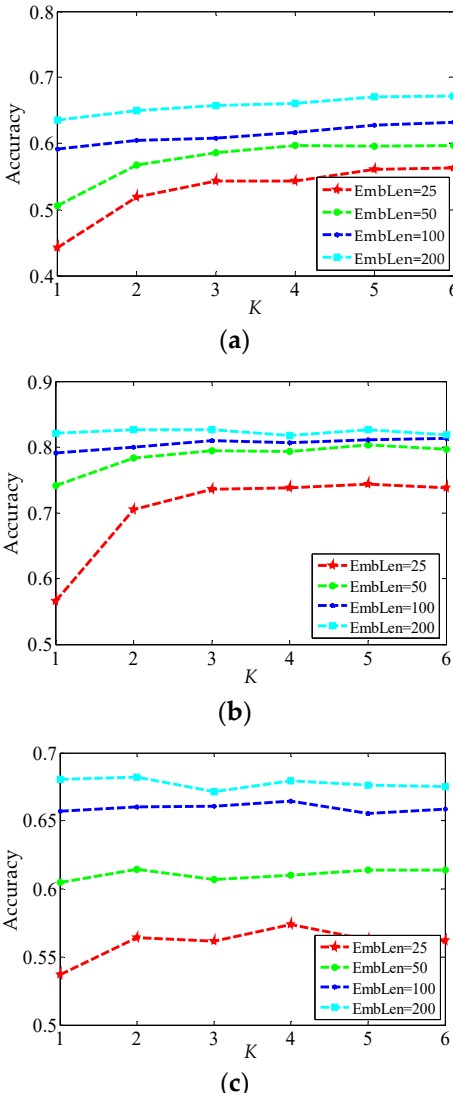

**Figure 2.** Parameter sensitivity analysis, here, we mainly research the influence of representation vector size and *K* value to network classification task. (**a**) parameter sensitivity analysis on Citeseer dataset; (**b**) parameter sensitivity analysis on Cora dataset; (**c**) parameter sensitivity analysis on Wiki dataset.

### 4.5. Network Visualization

In this experiment, we aim at visualizing the learnt representations on a real-world citation network (Citeseer dataset). We adopt the t-SNE [36] toolkit to visualize the learnt representations. First, we randomly sample 4 categories on Citeseer. Each category consists of 150 nodes generated by random sample approach. Based on this experiment, we aim to verify whether MVNR is quite qualified for learning discriminative representations. We visualize the learnt representation trained from DeepWalk, node2vec and MVNR algorithms. The representation length of MVRN is 600 and representation size of DeepWalk and node2vec is 200. the results of network representation are shown in Figure 3.

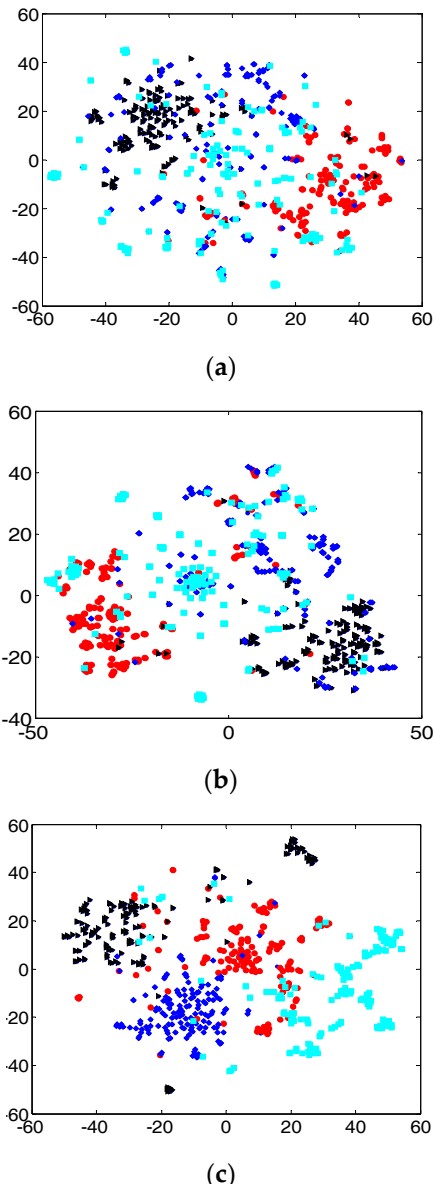

**Figure 3.** 2D Visualization on Citeseer. (**a**) Citeseer visualization using DeepWalk; (**b**) Citeseer visualization using node2vec; (**c**) Citeseer visualization using MVNR.

As shown in Figure 3, data points of different colors represent different kinds of nodes. We visualize the learnt network representations. It can be found, based on the visualization results, that the same color points tend to cluster together. The clustering ability of the network representations can reflect whether the trained model can embed the nodes with similar attributes and features into the representation space of closer space distance. For example, the similarity between nodes with links should be greater than that between nodes without links. In addition, the network representation with better clustering ability is helpful to the classification tasks of network nodes by embedding the nodes in the same category into the representation space of closer space distance.

Note that the Wiki dataset consists of 2405 nodes and 19 different categories. There are 126 nodes on average in each category and the number of nodes in most categories is less than 150, so we randomly select four labels in 19 categories, such that there exist 150 nodes at least in each category. Therefore, in Citeseer dataset, we also randomly selected 150 nodes for four categories. That is the reason why we choose four categories and 600 vertices on Citeseer. Random selection strategy can better reflect the learning ability of the proposed MVNE algorithm. In addition, the learnt network

representations are projected onto a two-dimensional plane, so we only select four categories of nodes in each dataset for clearly achieving clustering boundaries between different categories of nodes and avoiding the poor display performance caused by too many categories.

*4.6. Link Prediction*

The link prediction algorithm can predict the future link probabilities of non-existing edges between vertices. First, we need to give a reasonable score for each vertex pair. We see $r_i$ and $r_j$ as the representations of the vertex $i$ and vertex $j$, we calculate the similarity score based on the cosine similarity $(r_i \cdot r_j)/(||r_i||_2 \cdot ||r_j||_2)$. We use AUC to evaluate the performance of link prediction. We remove 30%, 20% and 10% edges of Citeseer, Cora and Wiki as test set and we use the remaining edges to learn the network representations. We also use some baseline algorithms of link prediction to make some comparison analysis with MVNR proposed in this paper. We set the *K* value as 3 and 6. The results are shown in Table 6.

**Table 6.** Link prediction performance results on Citeseer dataset.

| Dataset Name | Training Ratio | Algorithm Name | | | | | | | | | |
|---|---|---|---|---|---|---|---|---|---|---|---|
| | | CN | LP | RA | ACT | LHNII | Cos+ | LRW | SRW | MVNR (*K* = 3) | MVNR (*K* = 6) |
| Citeseer | 0.7 | 68.13 | 81.06 | 66.37 | 75.88 | 95.76 | 88.57 | 87.21 | 86.34 | 95.15 | 93.96 |
| | 0.8 | 72.08 | 86.83 | 72.12 | 75.59 | 96.85 | 89.38 | 90.13 | 90.05 | 95.70 | 96.22 |
| | 0.9 | 74.67 | 88.45 | 74.63 | 73.79 | 96.20 | 88.49 | 91.25 | 90.47 | 97.08 | 97.01 |
| Cora | 0.7 | 69.50 | 80.12 | 69.47 | 74.11 | 89.41 | 90.25 | 88.48 | 88.40 | 92.09 | 92.60 |
| | 0.8 | 72.38 | 82.97 | 72.47 | 73.67 | 90.37 | 90.98 | 90.58 | 90.50 | 93.15 | 92.47 |
| | 0.9 | 78.19 | 87.90 | 77.97 | 74.00 | 93.64 | 93.22 | 93.63 | 93.62 | 94.42 | 93.43 |
| Wiki | 0.7 | 85.79 | 92.60 | 85.99 | 80.19 | 86.45 | 91.49 | 93.65 | 93.48 | 93.38 | 93.66 |
| | 0.8 | 87.95 | 93.56 | 88.24 | 80.35 | 86.77 | 91.63 | 94.20 | 94.18 | 93.95 | 93.44 |
| | 0.9 | 90.47 | 94.00 | 90.33 | 79.98 | 87.68 | 92.71 | 94.56 | 94.53 | 94.65 | 94.60 |

As shown in Table 6, The MVNR algorithm proposed in this paper is compared with the 8 kinds of link prediction algorithms presented in Table 6. The experimental results show that the proposed MVNR algorithm always achieves the better performance under the three kinds of training proportions on Citeseer, Cora and Wiki datasets.

**5. Conclusions**

In this paper, we propose a unified network representation learning framework (MVNR) which can embed hider order neighboring relations and global information into the representation space of a lower dimension. In addition, we adopt the weight matrix factor to balance different *k*-step network representations with the modeling procedure of the joint representation learning, namely, we adopt the inductive matrix completion algorithm for synchronously learning the network features from both the network structures and link weights. Empirically, the learnt representations can be effectively applied to various machine learning tasks, such as clustering, classification, visualization and link prediction and so forth. The experimental results show that the proposed MVNR can effectively capture the global features and higher order relations between vertices at the same time. Meanwhile, the classification performance of MVRN outperforms the popular global network representation learning algorithms (Line) and higher order network representation learning algorithms (GraRep and NEU). Our Future works would explore the extendibility of our algorithm in other representation learning tasks.

**Author Contributions:** Conceptualization, Z.Y., K.Z. and Y.Z.; Methodology, Z.Y.; Software, K.Z.; Validation, Z.Y. and Y.Z.; Formal Analysis, Y.Z.; Investigation, Z.Y.; Resources, Z.Y.; Data Curation, K.Z.; Writing-Original Draft Preparation, Z.Y.; Writing-Review & Editing, Y.Z.; Visualization, Z.Y.; Supervision, H.Z.; Project Administration, H.Z.; Funding Acquisition, H.Z.

**Funding:** This research was supported by the National Natural Science Foundation of China (Grant Number: 61763041, 11661069 and 61663041), the Program for Changjiang Scholars and Innovative Research Team in Universities (Grant Number: IRT_15R40) and the Fundamental Research Funds for the Central Universities, China (Grant Number: 2017TS045).

**Acknowledgments:** The authors wish to express their gratitude to the anonymous reviewers and the associate editors for their rigorous comments during the review process. In addition, the authors also would like to thank experimenters in our laboratory for their great contribution to data-collection work. They are Ke Zhang and Yu Zhu.

**Conflicts of Interest:** The authors declare no conflict of interest.

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
