# Peer review of "Multi-View Network Representation Learning Algorithm Research"

_algorithms, doi:10.3390/a12030062_

Reviewer 1 Report

This paper presents MVNR - MultiView Network Representation algorithm, for generating graph embeddings on weighted graphs. The authors propose attaching weights to the different k-step networks of the original graph, and then using the IMC algorithm to find a suitable factorization of the network structural features matrix M, to get, for each step k in the range considered, R(k) - the representation matrix of the k-step network; then, by concatenating R(1), R(2)...R(k) - we get the final graph embedding.

While the approach is quite interesting, the paper is not sufficiently clearly written, the research choices are poorly motivated and are explained in a rather complicated fashion. Information belonging to different parts are mixed in the same section, which makes the paper difficult to read and the contribution difficult to assess. 

Specifically:

 - too many technical details in the intro - comparison with other similar methods should pe performed AFTER the method has been introduced (you present disadvantages of other methods in contrast with MVNR, before explaining either MVNR or presenting related the methods). 

 - too many details about related work in presenting the contribution - section 3 (perhaps consider adding a discussion section after the evaluation section, in which you discuss the differences to existing methods and why MVNR is better; move the part of the intro that provides part of the discussion there as well).

 In section 4.2. Baseline Algorithms, you shouldn't describe the algorithms used. Better explain why you have selected them, and give the corresponding parameter values used in the evaluations.

- the Matrix Forest Index is not suitably explained - why that particular choice for S? How does that balance the representation? Also, to me it is not clear how MFI is calculated. The authors say it is on a 1-step network (isn't it on the k-step reachability network?). Then how do you have several edges between two nodes? ("wxy is the link weight of p-th edge from vertex vx to vertex vy") You should clarify this.   

As you should also clarify why matrix M, as you define it, acurately captures correctly the network structural features, and why factorizing it gives you an information rich embedding. (you base your idea on TADW, but the matrix A you use is not the same as the one used in TADW - so, how are the findings in [29] valid? You should discuss this apect also). And also clarify "but we perform some optimization operations for the matrix A(k)" (which goes actually in line with better explaining you research/alg. design choices).

In the introduction, you state that "the link weights consist of the certainty degree of existing edges and the link probability of non-existing edges. The certainty degree of existing edges is applied to 1-step network, and the link probability of non-existing edges is used to k-step networks where k > 2.". But in section 3, where you present the contribution, this distinction is not clear. Moreover, I find sections 3.2 and 3.3 extremely convoluted and difficult to assimilate. Many formulations are of the form "smth is NOT like smth else" - you should instead make very clear what smth IS, and WHY you chose it to be like that - what are the benefits (richer information, increased speed, etc).

Regarding the complexity analysis, the paper states that: "In matrix M, the number of rows equals the number of columns, meanwhile, the number of rows and columns equals the number of vertices in the network, which is defined as |V|. Thus, the time complexity of constructing matrix M is O(|V|^2)." However, isn't matrix M obtained by raising matrix A to the power of 2? Matrix multiplication can be done in at least ~ O(|V|^2.8), if you use Strassen's multiplication algorithm (or O(|V|^3) if you use the traditional algorithm). Am I missing something?

The fact that the paper compares the algorithm with other prominent algorithms is a plus. In Table 2, does Category mean number of classes? AveragePath is the average path length? (why is it longer on a denser network)? Are the networks weighted, or not? The evaluation scenarios are appropriately chosen, results are reasonably interpreted. 

Specific observations:

- "In addition, ... is not only applied network classification task, but also can be used for various machine learning tasks, such as clustering, link prediction, recommendation systems etc." - why have such a sentence in the notation part of section 3? Also, "not only applied TO ... ".

- "A is not a adjacency matrix." -> "A is not an adjacency matrix." - also, since you give the value for any aij - it is not necessary to restate that A is not the adjacency matrix.

- "TADW [29] presents a simplified target matrix, and find a balance between" -> "TADW [29] presents a simplified target matrix, and finds a balance between"

- " In MVNR algorithm, we also" ->  "In MVNR , we also" or "MVNR also captures ... "

-"Matrix A(k) can be formulated as follows: A(k).... where A(k) is the adjacency matrix of the k-step network, a(k) is the matric elements of matrix A(k) , ij and adjacency matrix A consists of 0 and 1, so we set the element value as 0 or 1 in ...., where aij is the matrix elements of matrix A ." This is extremely convoluted! From what I gather, matrix A(k) is simply a k-step reachability matrix (i.e. boolean instead of "probability of reachability", as in A^k - PageRank). You should opt for a simpler explanation.

- "We fix the training ration to 0.5" -> "We fix the training ratio to 0.5"

-etc. (many articles missing, several reformulations needed to improve clarity and language quality, etc.)

Author Response

Dear Reviewer

Please see the Responses file in the attachment.

Best regards,

Zhonglin Ye

Reviewer 2 Report

This paper deals with the network embedding problem, and proposes a multi-view network representation (MVNR) method to embed the network nodes into some lower-dimensional space. Multiple steps of random walks are used to incorporate the higher-order information, where each step of random walk results in a network and differently steps of random walks are assigned different weights. Experiments are performed on several datasets to test the performance of MVNR. About this paper, I have several comments.

1. The motivation should be improved. There already have been many network embedding methods with higher-order relationship incorporated. The authors should give a clearer discussion what are the main advantages of the proposed method when compared to the existing network embedding methods, such DeepWalk, LINE, node2vec, etc.

2. In Line 54, page 2, it is stated that “However, DeepWalk [1] cannot capture the network structural features of higher order.”, which is not true. DeepWalk is able to incorporate higher-order network information by multiple steps of random walks. The discussion here should be improved.

3. In the end of Section 1, the contributions should be re-organized. Currently, the three contributions summarized by the authors are in fact the same one with different descriptions. As stated by the authors, the 1st contribution is the new MVNR method with higher-order and global relationship incorporated, the 2nd contribution is the multi-view strategy in MVNR to capture higher-order information, and the 3rd contribution is the muti-view strategy in MVNR makes full use of the structure information. In fact, these three contributions mean the same thing, which should be re-organized with better summarization.

4. The experiments should be improved. In lines 92, page 3, the authors stated that “For the final network 92 representations, the different k-step networks should be given different weights”. To prove this, the authors should conduct experiments to show how the weighting strategy improves the proposed MVNR method. Specifically, experimental comparison between MVNR with different weights (for k-step networks) and MVNR with the equal weights should be given, in order to make the weighting strategy convincing.

Author Response

Dear Reviewer

Please see the Responses file in the attachment.

Best regards,

Zhonglin Ye

Round  2

Reviewer 1 Report

The authors have addresed the major obsevations and recommendations provided. Some parts of the paper could still be improved (e.g.) :

- sect. 4.2 on baseline algorithms, the authors present both a list of the algorithms (brief description and parameter settings), but also added a paragraph after motivating the choice of algorithms - I would perhaps move the parameter settings for the selected baseline models in a table, and keep only the newly added paragraph as main text in this section.

- in the algorithm pseudocode, point 2 contains some unclear info - what do "eye", "logical" procedures perform? (sparse, sum and size are easy to grasp, but these 2 need further explaining)

Minor text editing and reformulations needed (e.g. " which are not weighted network. " - must use plural - "networks"; "As we mentioned as above,"; etc).

Author Response

Dear reviewers

The detailed descriptions for revision can be found in the attachment.

Best regards,

Zhonglin Ye

Reviewer 2 Report

The authors have addressed my concerns in the first round review.

Author Response

Dear Editors and Reviewers:

Question

The authors have addressed my concerns in the first round review.

Responses:

Thank you very much for the reviewers’ comments concerning our manuscript entitled “Multi-View Network Representation Learning Algorithm Research”. Your comments are all valuable and very helpful for revising and improving our paper, as well as the important guiding significance to our future researches.

Sincerely yours,

Zhonglin Ye